# *In vitro* one-pot construction of influenza viral genomes for virus particle synthesis based on reverse genetics system

Ryota Tanaka[1], Kenji Tamao[1], Mana Ono[1], Seiya Yamayoshi[2,3], Yoshihiro Kawaoka[2,3,4,5], Masayuki Su'etsugu[6], Hiroyuki Noji[1], Kazuhito V. Tabata[1] *

**1** Department of Applied Chemistry, Graduate School of Engineering, the University of Tokyo, Tokyo, Japan, **2** Division of Virology, Institute of Medical Science, University of Tokyo, Tokyo, Japan, **3** The Research Center for Global Viral Diseases, National Center for Global Health and Medicine Research Institute, Tokyo, Japan, **4** Department of Pathobiological Sciences, Influenza Research Institute, School of Veterinary Medicine, University of Wisconsin, Madison, Wisconsin, United States of America, **5** Pandemic Preparedness, Infection and Advanced Research Center (UTOPIA), University of Tokyo, Tokyo, Japan, **6** Department of Life Science, College of Science, Rikkyo University, Tokyo, Japan

* tabatak@g.ecc.u-tokyo.ac.jp

**Data Availability Statement:** All the data for this paper can be found in the article and in the repository. In particular, we have uploaded the

## Abstract

The reverse genetics system, which allows the generation of influenza viruses from plasmids encoding viral genome, is a powerful tool for basic research on viral infection mechanisms and application research such as vaccine development. However, conventional plasmid construction using *Escherichia coli* (*E.coli*) cloning is time-consuming and has difficulties handling DNA encoding genes toxic for *E.coli* or highly repeated sequences. These limitations hamper rapid virus synthesis. In this study, we establish a very rapid in vitro one-pot plasmid construction (IVOC) based virus synthesis. This method dramatically reduced the time for genome plasmid construction, which was used for virus synthesis, from several days or more to about 8 hours. Moreover, infectious viruses could be synthesized with a similar yield to the conventional *E.coli* cloning-based method with high accuracy. The applicability of this method was also demonstrated by the generation of recombinant viruses carrying reporter genes from the IVOC products. This method enables the pathogenicity analysis and vaccine development using genetically modified viruses, and it is expected to allow for faster analysis of newly emerging variants than ever before. Furthermore, its application to other RNA viruses is also expected.

## Introduction

Influenza is known as the seasonal cold/flu that causes epidemics and pandemics [1–3]. This is due to the evolution of the influenza viral genome through the introduction of mutations within the host [4, 5] and reassortment [6, 7]. Whenever influenza viruses with antigens that humanity has never experienced emerge in human society, it has led to global pandemics [8]. In particular, influenza A viruses caused three pandemics in the 20th century [9], and in the 21st century, there was a pandemic caused by a swine-origin influenza A virus in 2009 [10].

original electrophoresis results and sequence data in S3 Fig to the repository (https://osf.io/9fyeu/).

**Funding:** the Japan Science and Technology Agency for Core Research for Evolutional Science and Technology, JST CREST, Japan (JPMJCR18S6 and JPMJCR22N2), The Ogasawara Toshimasa Memorial Foundation. The funders had no role in study design, data collection and analysis, decision to publish, or preparation of the manuscript.

**Competing interests:** The authors have declared that no competing interests exist.

Considering the potential emergence of new pathogenic viruses [11], it is urgent to advance our biological understanding of the influenza A virus through research and the development of tools to counter its threat to humanity.

Recently, synthetic biology methods have been developed to synthesize virus particles to study viral infection mechanisms and develop vaccines [12–15]. In the field of influenza virus research, the reverse genetics system has been established. This system allows viruses to be generated from cDNA encoding the genomic viral RNA (vRNA) that is cloned into plasmids [16]. Specifically, eight genomic plasmids expressing each of the eight influenza vRNA genome segments and four genomic plasmids expressing each of the four influenza viral proteins necessary for viral genome replication are constructed and transfected into cultured cells to produce virus particles. This system has been used as an essential tool in current research on influenza viruses because it allows the generation of recombinant virus particles for various purposes by designing cDNA sequences [17–22].

In the reverse genetics system, it is essential to construct genomic plasmids containing cloned influenza viral cDNAs. Traditional *E.coli* cloning [23] presents several technical difficulties, while allowing for low-cost plasmid construction. Firstly, *E.coli* cloning requires many separate processes, such as transformation, culture, and colony picking, which can take several days to obtain the desired plasmid. Especially when multiple plasmids are required, as in reverse genetics system, these processes are required for each plasmid, resulting in more time and cost (Fig 1). In addition, it is difficult to handle DNA encoding genes toxic to *E.coli* or highly repetitive sequences prone to replication errors. When dealing with cDNA encoding the viral genome, cloning has been reported to be difficult due to the instability of the viral genome in *E.coli* [24–26]. These limitations reduce the throughput of virus particle synthesis using reverse genetics systems.

Several approaches have been published to reduce the effects of *E.coli*. One approach involves amplifying *in vitro* viral genomic plasmids using rolling circle amplification (RCA) and using that product for direct transfection into cells [27]. Although this approach is effective, complete cell-free genomic construction has not been achieved because the template plasmids used for RCA are constructed using *E.coli* cloning. In other RNA virus reverse genetics systems, several approaches have been developed that do not involve the use of *E.coli* at all. In the dengue virus (DENV) reverse genetics system, Gibson assembly (GA) is used [28]. In this

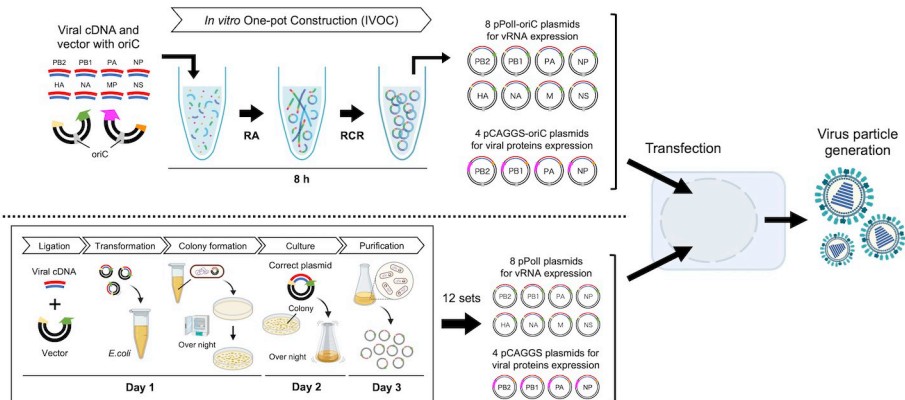

**Fig 1. Comparison between the IVOC-based method and the conventional method based on *E.coli* cloning.** The IVOC-based method (upper) requires only an 8-hour one-pot reaction for genomic plasmid construction, whereas the conventional method (lower) requires 12 construction steps that take 3 days per one plasmid.

system, the DENV genomic plasmid is reconstructed from multiple PCR products, which are amplified from viral cDNA, in a single GA reaction. The reaction products are then transfected directly into cells to produce the DENV. Recently, there have been reports of reverse genetics systems for flaviviruses and SARS-CoV-2 using Circular Polymerase Extension Reaction (CPER) [15, 29]. In these systems, multiple cDNA fragments covering the entire viral genome and DNA fragments encoding promoter and poly(A) signals are amplified and linked using PCR. The resulting genomic plasmids are then directly transfected into cells to synthesize the target virus. Although several approaches for synthesizing viruses using *in vitro* genome construction have been reported, none of the target RNA viruses have a segmented genome, requiring the construction of only one type of genomic plasmid. When dealing with multi-segmented genomes and the need to construct multiple types of genomic plasmids, such as in the case of influenza viruses, these approaches become more complex.

We focused on a recently developed technology called the RA-RCR system [30–32]. This system combines Recombination Assembly (RA) and Replication Cycle Reaction (RCR) for the cell-free construction of long-chain circular DNA. The RA system uses the RecA recombinase enzyme activity [33, 34] to assemble and circularize multiple DNA fragments *in vitro*. The RCR system reconstitutes the 25 proteins required for the *E.coli* DNA replication machinery *in vitro*. It exponentially amplifies circular DNA with the oriC sequence with a high accuracy of approximately $10^{-8}$/bp error rate per replication cycle [30]. This RA-RCR system requires fewer steps than traditional *E.coli* plasmid construction and can handle sequences with toxicity or repetitive sequences. The application of this technology to construct genomic plasmids in the reverse genetics system is expected to increase the throughput of influenza virus particle synthesis.

In this study, we developed a new method for the artificial synthesis of influenza virus particles to construct genomic plasmids in a one-pot reaction applying the RA-RCR system (Fig 1). This method allows the construction of all genomic plasmids required for the synthesis of influenza virus particles in one pot, and the reaction product is directly used for transfection to synthesize virus particles. This *in vitro* one-pot construction (IVOC) based method reduces the time required to construct genomic plasmids from several days to approximately 8 hours, enabling high-throughput virus particle synthesis. In addition, we demonstrated the feasibility of artificially synthesizing engineered recombinant influenza virus particles using this IVOC approach by generating recombinant virus particles carrying reporter genes.

## Materials and methods

### Cells and viruses

Madin-Darby Canine Kidney (MDCK) cells were obtained from the American Type Culture Collection, and human embryonic kidney (HEK293T) cells were obtained from RIKEN. MDCK and HEK293T cells were cultured at 37˚C with 5% $CO_2$ in Dulbecco's modified Eagle's medium (DMEM) supplemented with 10% fetal bovine serum (FBS) and 1% penicillin–streptomycin. Influenza virus (A/Puerto Rico/8/1934(H1N1)) was prepared as previously described [35]. Virus samples were stored at −80˚C and thawed and diluted when used.

### Construction of the viral genomic plasmids using the IVOC method

pPolI plasmids (pPolI-PR8-PB2, pPolI-PR8-PB1, pPolI-PR8-PA, pPolI-PR8-HA, pPolI-PR8-NP, pPolI-PR8-NA, pPolI-PR8-MP, and pPolI-PR8-NS) and pCAGGS plasmids (pCAGGS-PR8-PB2, pCAGGS-PR8-PB1, pCAGGS-PR8-PA, and pCAGGS-PR8-NP) were provided by Prof. Yoshihiro Kawaoka from Institute of Medical Science, University of Tokyo. pPolI plasmids was for the expression of the genomic vRNA segments of the PB2, PB1, PA,

HA, NP, NA, M, and NS of influenza A/PR/8/34 virus and pCAGGS plasmids was for the expression of PB2, PB1, PA, and NP proteins of influenza A/PR/8/34 virus. The vector fragments for vRNA expression and viral protein expression were generated by PCR using pPolI-oriC and pCAGGS-oriC plasmids as templates, which were created by inserting oriC sequences into the pPolI and pCAGGS plasmids, respectively. The cDNA fragments encoding the genomic vRNA were also generated by PCR using the pPolI plasmids pCAGGS plasmids as templates, designed to have 30–45 bp terminal overlap sequences with each vector fragment.

Plasmid construction using the RA-RCR system followed the protocol outlined in a previous report [30]. The RA mix and RCR mix required for the RA-RCR system were prepared by Su'etsugu et.al according to that report [30]. First, each fragment was quantified with Quantus™ fluorometer and then mixed in the RA mix in equimolar concentration, which incubated in assembly reaction at 42˚C for 30 minutes, followed by heat treatment at 65˚C for 2 minutes to eliminate the misassembled products. The RCR mix was primed by incubating at 33˚C for 15 minutes during the RA step. After priming, the RA product was mixed with the RCR mix and incubated at 33˚C for 6 hours to perform the plasmid amplification reaction and then held at 4˚C. The product was then supercoiled by diluting the product two-fold in $1 \times$ Amplification Buffer and incubating at 33˚C for 30 minutes. The product was then tested by PCR amplification specific for each genomic plasmid to confirm the proper assembly of eight pPolI-oriC plasmids and four pCAGGS-oriC plasmids in a one-pot reaction.

## Virus particle synthesis and quantification

Artificial synthesis of virus particles using the reverse genetics system was performed according to previous reports [12, 16, 36, 37]. First, the IVOC product containing all the genomic plasmids for virus particle synthesis was fully transfected into HEK293T cells (c cells) with the use of TransIT-293 Transfection Reagent (Mirus). 48 hours after transfection, Tosylsulfonyl phenylalanyl chloromethyl ketone treated trypsin (TPCK-Trypsin, Worthington Biochemical) was added (final conc. 1 ug/ml) and then incubated at 37˚C for 30 minutes with 5% $CO_2$. The culture supernatant was then collected and centrifuged at $1700 \times g$ for 10 minutes at 4˚C, and the supernatant was collected as virus samples. The synthesized virus particles were detected by adding virus samples to MDCK cells and observing cytopathic effects (CPEs). Virus particle quantification was performed using digital influenza assay [38] and plaque assay.

## Synthesis and detection of recombinant reporter virus particles

According to previous reports [39, 40], recombinant cDNA fragments encoding the influenza virus's hemagglutinin were designed by inserting the fluorescent protein mNeonGreen (mNG) gene. The genomic plasmids for synthesizing recombinant reporter viruses were then prepared from the DNA fragments using the IVOC method. The recombinant reporter virus particle synthesized from the IVOC product was infected with MDCK cells, and the expression of mNG in the cells was observed by fluorescence microscopy.

## Results and discussion

### Application of RA-RCR system for IVOC of viral genomic plasmids

First, we investigated the applicability of the RA-RCR system for *in vitro* one-pot construction (IVOC) of all twelve genomic plasmids for the synthesis of influenza virus particles. We used eight pPolI plasmids for vRNA expression and four pCAGGS plasmids for viral protein expression. Since DNA fragments containing oriC sequences are required for plasmid

**Fig 2. Verification of genomic plasmid construction for virus particle synthesis using the IVOC method.** (A) Verification by PCR of genomic plasmids constructed in the RA-RCR system. PCR amplification of the linkage of DNA fragments is performed by using primers that anneal to the viral cDNA site and the vector site comprising each plasmid (S1 and S2 Figs). (B) Confirmation of the construction of each genomic plasmid using agarose gel electrophoresis of the amplified products. Original images can be found in S1 Raw image.

construction using the RA-RCR system, pPolI-oriC vectors for vRNA expression and pCAGGS-oriC vectors for viral protein expression were prepared by PCR using pPolI-oriC and pCAGGS-oriC plasmids as templates, which were created by inserting oriC sequences into the pPolI and pCAGGS plasmids, respectively. In addition, viral cDNAs encoding each of the eight viral genomes and having complementary ends overlapping 30 to 45 nucleotides at the end of each vector were prepared using PCR.

The DNA fragments for the construction of the eight vRNA expression plasmids (pPolI-oriC vector and eight viral cDNAs) and for the construction of the four viral protein expression plasmids (pCAGGS-oriC vector and four viral cDNAs) were then mixed in equimolar concentrations for the paired combinations. IVOC of all twelve genomic plasmids was then performed in an eight-hour reaction using the RA-RCR system. After the reaction, specific PCRs were performed using each plasmid as a template (Fig 2A) to confirm that eight pPolI-oriC plasmids and four pCAGGS-oriC plasmids had been constructed in the obtained one-pot reaction product. As a result, amplicons of the correct size were detected in all PCRs (Fig 2B). These results showed that all genomic plasmids for virus particle synthesis were successfully constructed in an *in vitro* one-pot reaction in approximately 8 hours using the RA-RCR system.

## Virus particle synthesis using the IVOC approach

Next, we investigated virus particle synthesis using the IVOC approach, as IVOC using the RA-RCR system was confirmed to construct all genomic plasmids for virus particle synthesis. Twelve genomic plasmids (eight pPolI-oriC and four pCAGGS-oriC) for virus particle synthesis were prepared using the IVOC approach with each vector and viral cDNA fragment. The resulting IVOC products were transfected into HEK293T cells ($1 \times 10^6$ cells). At 3 days post-transfection (dpt), the culture supernatants of HEK293T cells were collected and added to MDCK cells for culture to verify whether infectious viruses were synthesized from the IVOC products. Fig 3 shows the images of MDCK cells before and after incubation at 35°C for 96 hours with the culture supernatants of HEK293T cells at 3 dpt. When the virus particles infect the cells and start to replicate inside the cells, cytopathic effects (CPE) such as rounding and detachment of the cells, appear in the infected cells. In the negative control, where MDCK cells were not infected with cultured influenza virus, no CPEs appeared in the cells (Fig 3A). In the positive control, where the cells were infected, CPEs such as rounding, detaching, and floating from the bottom of the culture dish were observed (Fig 3B). When the culture supernatants of HEK293T cells were added, CPEs were observed as in the positive control (Fig 3C). These

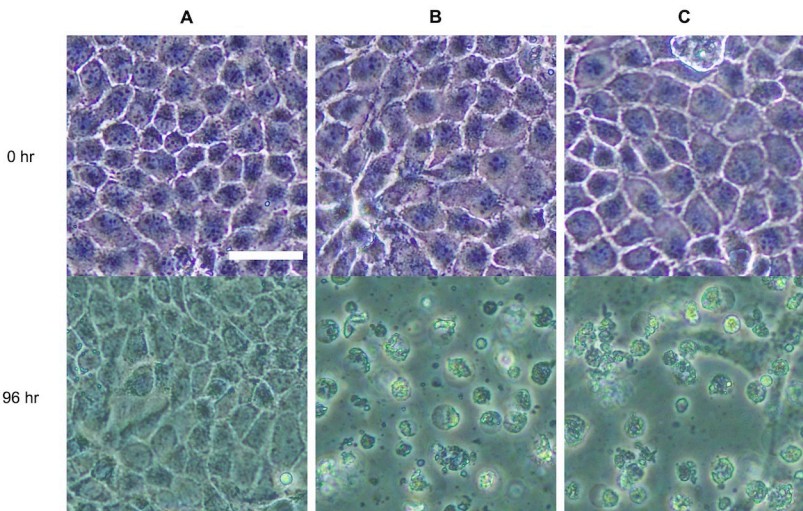

**Fig 3. CPE assay using MDCK cells.** (A) Negative control. MDCK cells are not infected with influenza viruses. (B) Positive control. MDCK cells infected with cultured influenza viruses. (C) MDCK cells with the culture supernatant of HEK293T cells (3 dpt) obtained by the IVOC-based method. The scale bar is 50 μm.

results showed that infectious influenza virus particles were contained in the culture supernatants of HEK293T cells at 3 dpt, strongly suggesting the IVOC-based virus particle synthesis.

In addition, to determine the full-length genome sequences of the viruses synthesized by the IVOC approach, the viruses obtained from the culture supernatant of HEK293T cells at 3 dpt were subjected to Sanger sequence analysis with specific primers. The analysis showed that the genome sequences of the synthesized viruses were identical to the viral genome sequences encoded in the transfected plasmid (S3 Fig), indicating that there were no significant mutations introduced during the construction and amplification of the viral genomic plasmid in the IVOC process.

## Quantification of virus particles synthesized using the IVOC approach

Next, we determined the number of synthesized virus particles by digital influenza assay [38] using the culture supernatants of HEK293T cells at the 3 dpt. Following a previous report [38], a femtoliter reactor array device was prepared, and virus particles were detected from the fluorogenic reaction between the fluorogenic substrate (2'-(4-methylumbelliferyl)-$\alpha$-D-N-acetylneuraminic acid, 4MUNANA) and the neuraminidase of the influenza virus particle in the reactors. The number of virus particles was determined from the signals from the virus particles synthesized using the IVOC-based method and compared with the conventional *E. coli* cloning-based method (Fig 4A). As a result, the number of virus particles synthesized using the IVOC-based method was $5.1(\pm 3.0) \times 10^8$ virus particles (VP)/mL, which was the same level as the conventional method ($4.4(\pm 0.8) \times 10^8$ VP/mL) (Fig 4C). The infectious titer of synthesized viruses was determined by plaque assay using MDCK cells (Fig 4B). As a result, the infectious titer of the virus particles synthesized using the IVOC-based method was $1.9(\pm 1.1) \times 10^2$ PFU/mL, which was approximately 1/20 of the titer using the conventional method ($4.6(\pm 4.1) \times 10^3$ VP/mL) (Fig 4C). Considering that virus samples are usually used in dilution series, this infectious titer was sufficient for practical use. These results showed that this IVOC-based method can reduce the time and process of viral genomic plasmids

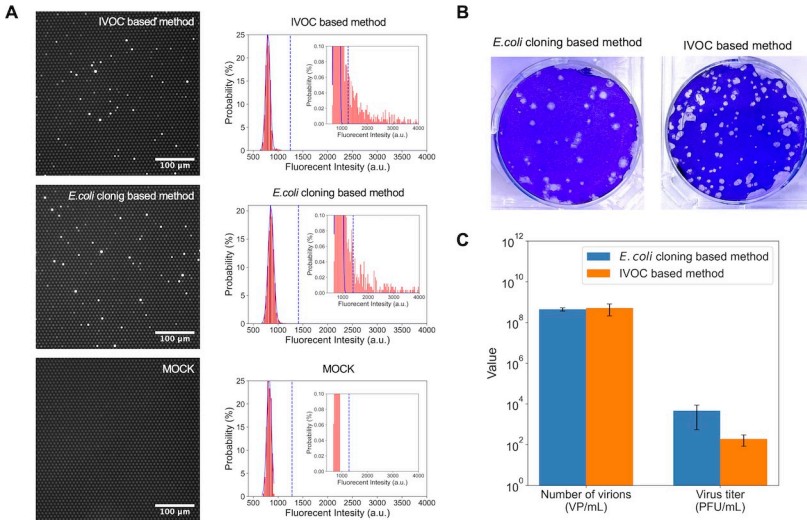

**Fig 4. Quantification of virus particles synthesized using the IVOC-based method and the conventional method.**
(A) The results of digital influenza assay. Left: Representative fluorescence images obtained using the culture supernatant of HEK293T cells (3 dpt). A single virus particle was encapsulated in a brightly illuminated chamber. Right: Fluorescence intensity distribution of the reactors. The blue lines show the normal distribution fits to the first peak corresponding to the empty reactors. The dotted lines show the threshold lines (mean + 10 × SD) to discriminate positive reactors from empty reactors. (B) The results of the plaque assay. (C) The results of the quantification of virus particles and infectious titers. The error bars represent the standard deviation.

construction, which previously took several days or more, to approximately 8 hours and generate sufficient quantities of infectious virus particles for practical use.

The lower infectious titer of the viruses synthesized using the IVOC-based method compared with the conventional method, even though the number of particles was at the same level, seems to be because the amount of transfection of each genomic plasmid has not yet been optimized. In the conventional method [16], each amount of transfection into cells can be optimized because genomic plasmids constructed using *E.coli* cloning are used. However, in the IVOC-based method, all the genomic plasmids are prepared by a one-pot construction reaction, and the IVOC products are directly transfected into cells for virus particle synthesis, so the transfection of each plasmid cannot be optimized.

The ratio of infectious titer to number of particles (PFU/VP) of viruses synthesized using the conventional and IVOC-based methods was approximately $10^{-5}$ and $10^{-6}$, respectively, which are lower than the values ($10^{-2}$ to $10^{-4}$) reported for cultured influenza viruses in previous studies [41, 42]. Recent studies have shown the presence of incomplete virus particles in influenza virus populations, which lack some of the eight genomic vRNA segments and are not infectious [43, 44]. Considering these findings, it is possible that similar phenomena affect the virus particles synthesized by transfection with genomic plasmids constructed using the reverse genetics system. The twelve genomic plasmids are required to synthesize infectious virus particles, and all of these plasmids must be introduced into a single cell to form complete virus particles that contain all the eight vRNA segments. However, the probability of this case is simply calculated to be as low as the twelfth power of $P$ ($0 < P < 1$), the probability of one plasmid being introduced. Therefore, it appears that most cells were not transfected with all twelve genomic plasmids, resulting in the generation of incomplete virus particles that lack some vRNA segments. Incomplete virus particles have been reported to be generated even when the number of genomic plasmids used for transfection was reduced [20]. These

phenomena suggest that the diversity of genomic plasmids for transfection contributes to the low PFU/VP ratio observed in influenza virus particle populations synthesized using the reverse genetics system.

## The synthesis of recombinant virus particles using the IVOC approach

Finally, we applied the IVOC approach to the synthesis of recombinant influenza virus particles carrying reporter genes. We attempted to artificially synthesize a recombinant virus particle carrying the gene for the fluorescent protein mNeonGreen (mNG) in the hemagglutinin (HA) vRNA segment and express fluorescence in infected cells. The recombinant HA vRNA was designed by inserting the mNG gene and a linker sequence into 78 nucleotides of HA vRNA (Fig 5A). The recombinant HA cDNA fragment was prepared by subjecting the vRNA to RT-PCR as templates. Then, 13 genomic plasmids for mNG-carrying virus (mNG virus) particle synthesis, including another pCAGGS-oriC plasmid for HA protein expression, were prepared using the IVOC method based on previous reports [39, 40]. The IVOC products were transfected into HEK293T cells ($1.9(\pm1.1) \times 10^2$ cells), and the culture supernatants at 3 dpt were collected as passage (P) 0 mNG viruses (Fig 5B). Fig 5C shows the results of quantifying the number of the mNG virus particles (P0) by digital influenza assay. The signals detected indicated the synthesis of $8.5 \times 10^7$ VP/mL mNG virus particles. We also examined the expression of mNG in MDCK cells after infection with the synthesized mNG virus particles (P0) and

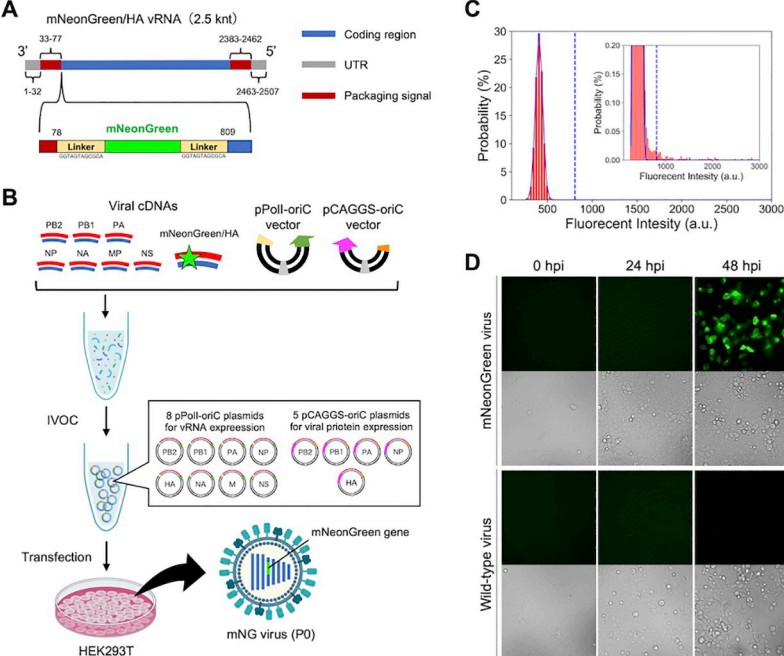

**Fig 5. Artificial synthesis of recombinant influenza virus particles carrying reporter genes using the IVOC-based method.** (A) Gene structure of the HA vRNA segment carrying the mNG gene. (B) Workflow of mNG virus particle synthesis using the IVOC-based method. 13 genomic plasmids for mNG virus particle synthesis were generated using the IVOC method from DNA fragments containing cDNA fragments encoding the recombinant HA vRNA and transfected into HEK293T cells. (C) Quantification of mNG virus particles (P0) by digital influenza assay. The number of mNG virus particles was determined from the fluorescence intensity distribution of the reactors. The blue lines show the normal distribution fits to the first peak corresponding to the empty reactors. The dotted lines show the threshold lines (mean + 10 × SD) to discriminate positive reactors from empty reactors. (D) Fluorescence signals in MDCK cells infected with WT and mNG virus particles.

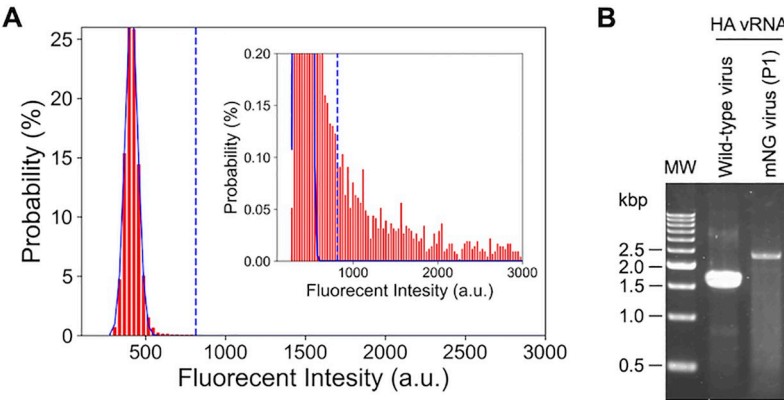

**Fig 6. Verification of progeny virus particles of mNG viruses synthesized using the IVOC-based method.** (A) Quantification of the mNG virus particles (P1) by digital influenza assay. The number of P1 virus particles was determined from the fluorescence intensity distribution of the reactors. The blue lines show the normal distribution fits to the first peak corresponding to the empty reactors. The dotted lines show the threshold lines (mean + 10 × SD) to discriminate positive reactors from empty reactors. (B) The results of RT-PCR of HA vRNA from the WT and the P1 virus particles. Original images can be found in S1 Raw image.

wild-type virus particles (WT virus) up to 48 hours post-infection (hpi) by fluorescence microscopy. The number of mNG-positive cells was observed at 48 hpi (Fig 5D).

Propagation of the progeny virus particles was examined by infection of MDCK cells with mNG virus particles (P0). The culture supernatants of MDCK cells at 2 dpi with the mNG virus (P0) were collected as the P1 viruses and quantified using digital influenza assay. As a result, signals from the P1 virus particles were detected, and the number of virus particles was determined to be $1.6 \times 10^9$ VP/mL (Fig 6A). In addition, genomic vRNA was extracted and purified from the P1 virus, and the length of the HA vRNA of the P1 virus was verified using RT-PCR. The results showed that the HA vRNA of the P1 virus was larger by the size of mNG gene ($\approx 0.7$ kbp) [45] than that of the WT virus (Fig 6B). These results indicated that recombinant influenza viruses carrying reporter genes could be rapidly constructed using the IVOC-based method.

## Conclusion

The reverse genetics system is one of the essential tools for influenza virus research. However, the low throughput of genomic plasmid construction in the conventional *E.coli* cloning-based method has hindered rapid virus particle synthesis. Here, we established a quick reverse genetics system for influenza A viruses based on the IVOC method. This system allows us to construct all the genomic plasmids (eight pPolI-oriC and four pCAGGS-oriC) required for virus particle synthesis *in vitro* using PCR and RA-RCR. Specifically, the construction of genomic plasmids, which previously took over three days, has been reduced to approximately 8 hours, and infectious virus particles can be artificially synthesized in high throughput.

The genome sequence analysis of the viruses synthesized using this method showed that there were no significant mutations introduced during the plasmid construction process, suggesting that this method had high accuracy. The applicability of this method was demonstrated by the generation of recombinant virus particles carrying reporter genes from the IVOC products. Because the RA-RCR system underlying this method can design and amplify circular DNA from two or more types of DNA fragments [30–32], improvements in this method are

expected to integrate the genomic plasmids required for virus particle synthesis, reduce the types of plasmids transfected into cells, and increase the efficiency of virus particle synthesis.

In this study, viral RNA isolated and purified from infectious cell culture supernatants or patient/animal samples was not tested as a template in the IVOC approach. However, it is anticipated that this would be feasible within the IVOC system, as viral RNA has successfully served as a template for virus production in the conventional methods. One potential application of the IVOC approach is the simultaneous generation of mutant libraries of influenza viruses using randomly mutated DNA fragments. This is expected to be used in situations such as screening for drug-resistant mutations. In addition, as the use of the reverse genetics system expands to viruses other than influenza viruses, the application of the IVOC approach to other RNA/DNA viruses can be expected.

## Supporting information

**S1 Fig. The detail information of primers used for verifying the construction of eight pPolI-oriC plasmids.** Each plasmid was constructed by assembling a vector fragment containing the oriC sequence, the human RNA polymerase I promoter (green, Pol I promoter), and the mouse RNA polymerase I terminator sequence (orange, Pol I terminator), together with a cDNA fragment (red) encoding each influenza viral segment in negative-sense orientation. PCR amplification was performed across the junction between the vector and the viral cDNA fragment using the primer pairs designed to bind to each fragment. "Fw" indicates the forward primer, and "Rv" indicates the reverse primer.
(PDF)

**S2 Fig. The detail information of primers used for verifying the construction of pCAGGS-oriC plasmids.** Each plasmid was constructed by assembling a vector fragment containing the oriC sequence, an RNA polymerase II promoter (pink, Pol II promoter, e.g., chicken *β*-actin promoter), a polyadenylation sequence (orange, PolyA, e.g., the rabbit *β*-globin polyadenylation sequence), and a gene sequence (black, chimeric intron) that includes introns from the chicken *β*-actin gene and rabbit *β*-globin gene, together with a cDNA fragment (red) encoding each viral protein. PCR amplification was performed across the junction between the vector and the viral cDNA fragment using the primer pairs designed to bind to each fragment. "Fw" indicates the forward primer, and "Rv" indicates the reverse primer.
(PDF)

**S3 Fig. Sanger sequencing of RT-PCR products from IVOC-synthesized viral genomic RNA.** The results are mapped to the reference, which is gene encoding region of the plasmids used for transfection, via GENETYX-NGS. The position of sequence primer is shown above each mapping result. The darker parts of each sequencing result indicate differences from the reference sequence. Sequencing results completely match the reference sequence except for the ends where the sequence accuracy is reduced. The following original data has been uploaded to the repository (https://osf.io/9fyeu/).
(PDF)

**S1 Raw image. Raw images for Figs 2B and 6B.** Agarose gel electrophoreses for verification of plasmid construction using the IVOC method (Raw Fig 2B) and demonstration of recombinant virus particle synthesis using the IVOC approach (Raw Fig 6B).
(PDF)

## Author Contributions

**Conceptualization:** Kazuhito V. Tabata.

**Formal analysis:** Ryota Tanaka, Kenji Tamao, Mana Ono.

**Funding acquisition:** Kazuhito V. Tabata.

**Investigation:** Ryota Tanaka, Kenji Tamao, Mana Ono.

**Methodology:** Seiya Yamayoshi, Yoshihiro Kawaoka, Masayuki Su'etsugu.

**Project administration:** Kazuhito V. Tabata.

**Resources:** Seiya Yamayoshi, Yoshihiro Kawaoka, Masayuki Su'etsugu, Hiroyuki Noji, Kazuhito V. Tabata.

**Supervision:** Kazuhito V. Tabata.

**Writing – original draft:** Ryota Tanaka.

**Writing – review & editing:** Hiroyuki Noji, Kazuhito V. Tabata.

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
