## [Decision Letter · Decision Letter 0]

21 Aug 2024

PONE-D-24-30909*In vitro* one-pot construction of influenza viral genomes for virus particle synthesis based on reverse genetics systemPLOS ONE

Dear Dr. Tabata,

Thank you for submitting your manuscript to PLOS ONE. After careful consideration, we feel that it has merit but does not fully meet PLOS ONE’s publication criteria as it currently stands. Therefore, we invite you to submit a revised version of the manuscript that addresses the points raised during the review process.

We look forward to receiving your revised manuscript.

Kind regards,

Herman Tse

Academic Editor

PLOS ONE

Journal Requirements:

"the Japan Science and Technology Agency for Core Research for Evolutional Science and Technology, JST CREST, Japan (JPMJCR18S6 and JPMJCR22N2), The Ogasawara Toshimasa Memorial Foundation"

3. Please note that funding information should not appear in the Acknowledgments section or other areas of your manuscript. We will only publish funding information present in the Funding Statement section of the online submission form. Please remove any funding-related text from the manuscript.

**Additional Editor Comments:**

The reviewer has pointed out a few minor issues, particularly with regards to some technical descriptions. Otherwise, this is a well-written manuscript which I would be ready to accept once the revisions are made.

Reviewers' comments:

Reviewer's Responses to Questions

**Comments to the Author**

1. Is the manuscript technically sound, and do the data support the conclusions?

Reviewer #1: Yes

2. Has the statistical analysis been performed appropriately and rigorously? 

Reviewer #1: N/A

3. Have the authors made all data underlying the findings in their manuscript fully available?

Reviewer #1: Yes

4. Is the manuscript presented in an intelligible fashion and written in standard English?

Reviewer #1: Yes

5. Review Comments to the Author

Reviewer #1: The manuscript by Tanaka et al entitled “In vitro one-pot construction of influenza viral genomes for virus particle synthesis based on reverse genetics system“ describes a novel approach to generate influenza A viruses from plasmid DNA. Using the in vitro one-pot plasmid construction (IVOC) reverse genetics approach, which exploits the E.coli DNA replication machinery in vitro, the authors show that they are able to generate all influenza A virus rescue plasmids within 8 hours instead of 3 days with conventional cloning techniques. Therefore, the IVOC approach offers the advantage to rapidly generate virus mutants that can be used for high throughput screens. The manuscript and the method described therein describe a novel cloning approach for the rescue of influenza A viruses and thus represent a valuable resource; however, there are some points listed below that should be addressed by the authors.

Specific comments:

In the abstract: It is unclear to the reviewer why the authors expect that this method “potentially advance further understanding of influenza viruses…”.

In the introduction: Please clarify that only influenza A viruses have caused known pandemics.

General comments:

The reviewer understood the IVOC approach to mean that the authors generated pPolI and pCAGGS plasmids with an E. coli oriC. For clarity, the authors should consider naming these modified plasmids, e.g. as pPolI-oriC, rather than referring to them as "these plasmids". In addition, Figure 1 should be adjusted accordingly to distinguish the plasmids with oriC (upper part of the figure) from the conventional pPolI and pCAGG plasmids without the oriC.

In the Materials and Methods section: Authors should provide the manufacturers of all reagents, especially the E. coli enzyme mix required for the IVOC method. Is it commercially available?

Because it is unclear to the reviewer why segments of similar length (e.g. PB2 and PB1) would generate a different amplicon size, the authors should provide a list of the primer pairs (and show were they bind in the plasmid backbone) used to generate the PCR products shown in Figure 2B.

In this study, the authors generated PR8 viruses using available plasmids as a template. The authors should comment on whether or not they tested isolated vRNA from infectious cell culture supernatant as a template for the IVOC approach. The authors should also mention whether or not this approach can be used to rapidly generate a plasmid rescue system for virus isolates from infectious patient/animal material.

6. PLOS authors have the option to publish the peer review history of their article (what does this mean?). If published, this will include your full peer review and any attached files.

Reviewer #1: No

---

## [Author Response · Author response to Decision Letter 0]

5 Oct 2024

Reviewer #1

Specific comments:

1. In the abstract: It is unclear to the reviewer why the authors expect that this method “potentially advance further understanding of influenza viruses…”.

At the end of the abstract, the following text was included to explain how the IVOC method can be used in influenza virus research.

This method enables the pathogenicity analysis and vaccine development using genetically modified viruses, and it is expected to allow for faster analysis of newly emerging variants than ever before. Furthermore, its application to other RNA viruses is also expected.

2. In the introduction: Please clarify that only influenza A viruses have caused known pandemics.

We have made the following changes to the introduction section you pointed out.

In particularthe 20th century, influenza A viruses caused there were 3 three pandemics in the 20th century (9), and in the 21st century, there was a pandemic was caused by a swine-origin influenza A virus in 2009(10). Considering the potential emergence of new pathogenic viruses(11), it is urgent to advance our biological understanding of the influenza A virus through research and the development of tools to counter its threat to humanity.

General comments:

1. The reviewer understood the IVOC approach to mean that the authors generated pPolI and pCAGGS plasmids with an E. coli oriC. For clarity, the authors should consider naming these modified plasmids, e.g. as pPolI-oriC, rather than referring to them as "these plasmids". In addition, Figure 1 should be adjusted accordingly to distinguish the plasmids with oriC (upper part of the figure) from the conventional pPolI and pCAGG plasmids without the oriC.

As you suggested, in order to make the names of the plasmids easier to understand, we have changed the plasmids used as templates for PCR to pPolI and pCAGGS, and the plasmids that have oriC to pPolI-oriC and pCAGGS-oriC. Therefore, we have made several corrections to the main text. In addition, we have made corrections to Figures 1 and 5 to distinguish between plasmids that do and do not have oriC. Please confirm the relevant sections.

2. In the Materials and Methods section: Authors should provide the manufacturers of all reagents, especially the E. coli enzyme mix required for the IVOC method. Is it commercially available?

All the reagents required for RA-RCR were prepared by Su’etsugu et al. The text has been modified as follows to make this clear.

Plasmid construction using the RA-RCR system followed the protocol outlined in a previous report(30). The RA mix and RCR mix required for the RA-RCR system were prepared by Su’etsugu et.al according to that report(30).

3. Because it is unclear to the reviewer why segments of similar length (e.g. PB2 and PB1) would generate a different amplicon size, the authors should provide a list of the primer pairs (and show were they bind in the plasmid backbone) used to generate the PCR products shown in Figure 2B.

The primer sets used in the PCR in Fig. 2B are listed in the Supporting Information as Fig. S1 and S2, along with the sequences and annealing positions for each plasmid. This is also shown in the legend for Fig. 2. As a result of this change, the previous Fig. S1 has been moved to Fig. S3.

4. In this study, the authors generated PR8 viruses using available plasmids as a template. The authors should comment on whether or not they tested isolated vRNA from infectious cell culture supernatant as a template for the IVOC approach. The authors should also mention whether or not this approach can be used to rapidly generate a plasmid rescue system for virus isolates from infectious patient/animal material.

In this study, we did not perform IVOC using vRNA isolated from clinical specimens. However, because general reverse genetics methods reconstitute viruses from vRNA derived from clinical specimens, we believe that IVOC is also possible. We will perform such experiments in the future. In this regard, we have also added the following text to the main text.

In this study, viral RNA isolated and purified from infectious cell culture supernatants or patient/animal samples was not tested as a template in the IVOC approach. However, it is anticipated that this would be feasible within the IVOC system, as viral RNA has successfully served as a template for virus production in the conventional methods.

---

## [Editor Report · Decision Letter 1]

14 Oct 2024

*In vitro* one-pot construction of influenza viral genomes for virus particle synthesis based on reverse genetics system

PONE-D-24-30909R1

Dear Dr. Tabata,

We’re pleased to inform you that your manuscript has been judged scientifically suitable for publication and will be formally accepted for publication once it meets all outstanding technical requirements.

Kind regards,

Herman Tse

Academic Editor

PLOS ONE
---

## [Editor Report · Acceptance letter]

28 Oct 2024

PONE-D-24-30909R1 

PLOS ONE

Dear Dr. Tabata, 

I'm pleased to inform you that your manuscript has been deemed suitable for publication in PLOS ONE. Congratulations! Your manuscript is now being handed over to our production team.

Kind regards, 

on behalf of

Dr. Herman Tse 

Academic Editor

PLOS ONE